# What Determines the Competitive Success of Young Croatian Wrestlers: Anthropometric Indices, Generic or Specific Fitness Performance?

**DOI:** 10.3390/jfmk8030090

**Published:** 2023-06-24

**Authors:** Krešo Škugor, Barbara Gilić, Hrvoje Karninčić, Matyas Jokai, Gergely Babszky, Marijana Ranisavljev, Valdemar Štajer, Roberto Roklicer, Patrik Drid

**Affiliations:** 1Faculty of Kinesiology, University of Split, 21000 Split, Croatia; kresoskugor95@gmail.com (K.Š.); barbara.gilic@kifst.eu (B.G.); 2Faculty of Physical Education and Sport Sciences (TF), Hungarian University of Sports Science, 1123 Budapest, Hungary; jokai.matyas@tf.hu (M.J.); babszky.gergely@gmail.com (G.B.); 3Hungarian Wrestling Academy “Kozma István”, 1213 Budapest, Hungary; 4Faculty of Sport and Physical Education, University of Novi Sad, 21000 Novi Sad, Serbia; marijanaranisavljev@gmail.com (M.R.); stajervaldemar@yahoo.com (V.Š.); roklicerr@gmail.com (R.R.); patrikdrid@gmail.com (P.D.); 5Wrestling Academy of Vojvodina, 24420 Kanjiza, Serbia; 6Faculty of Education, Free University of Bozen-Bolzano, Viale Ratisbona 16, 39042 Brixen-Bressanone, Italy

**Keywords:** combat, sports performance, sports success, talent identification, youth

## Abstract

Identifying factors influencing wrestling performance is important for determining which capacities should be developed the most. This research aimed to investigate whether anthropometric indices, generic fitness, and specific fitness performance determine the competitive success of young wrestlers. This research included 49 Croatian Greco–Roman male wrestlers aged 17.75 ± 1.51 years. Variables included training and competing experience, anthropometric indices, generic fitness (countermovement jump and handgrip strength), and specific wrestling fitness test (SWFT). Wrestlers were separated into medallists and non-medallists (i.e., wrestlers who won a medal at the previous National Championship and wrestlers who did not win a medal, respectively). The t-test for independent samples was used to determine the differences between the two categories in all variables. Moreover, discriminant analysis was performed to identify differences in a multivariate manner. Medallists and non-medallists did not differ in anthropometric indices and wrestling experience. Medallists had better results in the countermovement jump (t = 2.55, *p* < 0.01), handgrip strength (t = 2.77, *p* < 0.01), and SWFT performance (t = 2.29, *p* < 0.05) than non-medallists. The discriminant analysis confirmed that performance categories differed in generic and specific fitness tests (Wilks’ Lambda = 0.73, *p* < 0.05). It could be suggested that coaches should develop both the generic and specific fitness of their wrestlers to become more successful at competitions.

## 1. Introduction

The Olympic style of wrestling is an intermittent activity that, in its structure of fighting, requires highly developed physical capacities and technical and tactical abilities. The Olympic-style wrestling match consists of two 3-min rounds, with a 30 s break between the rounds [1]. The match is very dynamic, characterized by sudden attacks and counterattacks, and it is clear that aerobic and anaerobic metabolic energy systems are engaged during and after a wrestling match. During the match, the aerobic system supports continuous effort and recovery, while the metabolic system of anaerobic energy becomes critical in the movements that occur in the submaximal and maximal load zones [2]. It is well known that the anaerobic system provides a short and quick burst of maximum power. Having highly developed anaerobic capacities would lead to a better wrestling performance [3]. Indeed, it was estimated that, during a wrestling match, 90% of energy needs come from anaerobic energy pathways, which means that anaerobic glycolysis prevails in a match [4,5]. Furthermore, anaerobic capacity has been suggested as a capacity that can differentiate successful from less successful wrestlers [6]; and, as such, is the focus of diagnostic testing of wrestlers.

As previously stated, a wrestling match includes explosive attacks and counterattacks that enable a wrestler to throw an opponent, pin him to the mat or escape from critical bottom positions. All that effort requires a high level of muscular power in the lower limbs [6,7]. A study on Tunisian wrestlers showed the relationship between lower-body peak power, evaluated by the Wingate test, and performance on a specific wrestling test that included throwing a partner [8]. Moreover, elite wrestlers had a higher vertical jump height and power compared to amateur wrestlers, indicating that vertical jump height (i.e., power of the lower extremities) is essential for wrestling success [9]. Furthermore, upper body strength, especially handgrip strength (HGS), is extremely important during the wrestling match for pulling, pushing, throwing manoeuvres, and controlling the opponent [10]. HGS had a strong relationship with success in wrestling (ranking at the competition) and has been reported as an ability that differentiated successful from less successful junior and senior wrestlers [9,11].

In the wrestling community, coaches and scientists are interconnected with the common goal of developing and utilizing specific performance tests that will assess wrestlers’ anaerobic and aerobic capacities [12]. One recently developed test is the specific wrestling fitness test (SWFT) which is constructed to imitate the demands of the wrestling match [13,14]. The SWFT consists of the maximal number of suplex throws (i.e., bridging and slamming the opponents/dummies on their backs) during 3 periods of 30 s with 20 s of rest between the throwing rounds [13]. This test can be a useful tool for directly evaluating a wrestler’s physical and physiological readiness, their specific level of fitness, and it has been proven to differentiate wrestlers of different qualities, with the assumption that it assesses the anaerobic energy system [15]. However, a study by Markovic et al., 2022 [15], only evaluated differences according to specific tests. There was no attempt to compare the same group of tests with the tests that comprise the components of generic fitness. Moreover, a recent study on elite wrestlers did not record differences in SWFT performance between national team members and non-team members [16]. The authors believe that such findings were recorded because only elite athletes were included and suggest that future studies should also include lower-level wrestlers.

Generally, there is a low number of studies that investigated differences in fitness performances according to competitive success among young wrestlers, and such studies are especially lacking among Croatian wrestlers. Similarly, a few studies simultaneously investigated differences between quality groups in both generic and specific fitness tests in youth wrestlers. Thus, the aim of this study was to examine whether Croatian youth wrestlers differ in anthropometric indices and generic and specific fitness according to their performance (i.e., competitive) quality. We hypothesize that the more successful wrestlers would have favorable body indices and better fitness test results than their less successful peers, which means that the selected tests and assessment tools would be able to differentiate wrestlers according to their performance quality. The results of this research would help clarify the factors that differentiate successful from less successful youth wrestlers and further determine which capacities are important for success in wrestling. Moreover, this study could be used as a guideline for selecting tests and assessment tools that are appropriate and able to differentiate the performance quality of youth wrestlers. Simply put, results could aid in creating training programs that help less successful wrestlers reach their full potential and become more successful.

## 2. Materials and Methods

### 2.1. Participants

This research included 49 Croatian Greco–Roman male wrestlers aged 17.75 ± 1.51 years, allocated to cadet and junior competitive categories. Wrestlers were selected from different clubs in four Croatian cities. To be included in the research, wrestlers had to have at least three years of wrestling experience to be able to perform all the tests, with emphasis on the wrestling-specific test. Furthermore, wrestlers with any illness or injury that might have prevented them from performing maximal tests were excluded from this study. The required sample size was 37 wrestlers, which was calculated according to the total sample of 170 wrestlers (69 cadets and 101 juniors) that competed at a previous cadet and junior National Championship, meaning that a sufficient number of participants was included in this research. Wrestlers were divided into two quality/performance categories: medallists, wrestlers who won a medal at the National Championship (n = 26), and non-medallists who did not win a medal at the National Championship (n = 23). Wrestlers were informed about the aims and procedures of the research and signed an informed consent before the investigation began (legal guardians signed an informed consent for participants under 18 years of age). This study was approved by the Ethical Board of the Faculty of Kinesiology, University of Split (Ref. no. 2181-205-02-05-22-0012).

### 2.2. Variables

This study included anthropometric indices, body composition, generic fitness tests, and a specific wrestling test.

Anthropometric indices included body mass (BM), body height (BH), body mass index (BMI), and body composition, i.e., body fat percentage. Body mass index was calculated using the following formula: BMI = BM (kg)/BH (m)^2^. Body fat percentage was calculated using the Slaughter–Lohman formula, which includes the sum of the triceps and calf skinfolds, measured by a Harpenden skinfold calliper (British Indicators, Burgess Hill, England).

Generic fitness tests included HGS and the countermovement jump (CMJ). HGS was measured using the electronic hand dynamometer (Camry, Model EH101, Zhongshan Camry Electronic Co. Ltd., Kina). Wrestlers had three trials of maximum effort, with the arm adducted and elbow flexed at 90°. The best result (i.e., the highest number) was taken for further analysis.

CMJ was measured by the Optojump system (Microgate, Bolzano, Italy). Athletes had to stand between the two photoelectric beams with their hands on their hips. They were instructed to assume the starting position and conduct a maximal vertical jump. Wrestlers had three jumping trials with a 1-minute rest between the trials. The best result (i.e., the highest jump) was taken for further analysis.

A specific wrestling fitness test (SWFT) was used to determine sport-specific fitness. This test was recently developed by Markovic [13] and showed appropriate reliability and validity in wrestlers [14,15]. The participants throw a weighted dummy using the suplex technique, during three 30 s rounds performing a maximum possible number of throws, with 20 s of rest between the rounds. The weight of the dummy was adjusted according to weight categories: wrestlers who weigh 55–67 kg performed the test with a 23 kg dummy, wrestlers in the 72–87 kg category with a 25 kg dummy, and wrestlers weighing over 90 kg with a 30 kg dummy. Athletes had heart rate monitors (POLAR H10, Polar Inc., Lake Success, NY, USA) during the testing period and the rest period after the test. Heart rate was observed immediately after the test ended and after 1 minute of rest. The test results included the total number of throws and the index, which was calculated as the sum of heart rate values divided by the total number of throws, as previously proposed [17].

### 2.3. Testing Protocol

All testing procedures were conducted during the morning (to avoid diurnal variation) and by the same investigators. Testing was performed in a single day in the following order. Anthropometric indices were tested before initiating other physical testing procedures. After collecting anthropometric data, wrestlers completed a 15 min warm-up session which consisted of running, jumping, mobility exercises, and wrestling-specific exercises that all athletes were familiar with and conducted at almost every training session. After the warm-up, wrestlers performed the CMJ and HGS tests. Finally, they performed 10–15 throws to familiarize themselves with the SFWT, after which they had a 5 min break, followed by the SWFT.

### 2.4. Statistical Analysis

The normality of the variables was checked using the Kolmogorov–Smirnov test. Descriptive statistics included means and standard deviations for all quantitative variables. As all variables were normally distributed, the t-test for independent samples was used to determine the differences between the two categories (medallists vs. non-medallists) in all variables. Moreover, discriminant analysis was performed to identify quality differences in a multivariate manner or, more precisely, to identify variables that predicted membership in the quality group. All analyses were conducted using Statistica 13.5 (TIBCO, Palo Alto, CA, USA) and applying a *p*-level of 0.05.

## 3. Results

Descriptive statistics and differences between medallists and non-medallists in all variables are shown in Table 1. It is evident that wrestlers do not differ in anthropometric indices or wrestling experience. However, they vary in performance variables in both generic and sport-specific tests.

Figure 1 shows differences in the variables that significantly distinguish quality groups of wrestlers, calculated by one-way ANOVA. It can be observed that medallists have better results than non-medalists in all tests, and the HGS (Figure 1B) is the variable that discriminates medalists from non-medallists the most, with medallists having better results.

Results from the discriminant function analysis are presented in Table 2. The factor structure revealed that fitness variables (CMJ, HGS, and SWFT) almost equally account for classifying wrestlers into their performance groups, which was not the case for the anthropometric variables.

## 4. Discussion

This study investigated whether Croatian youth wrestlers differ in anthropometric indices and generic and specific fitness according to their competition performance. The main findings of this research are: (i) wrestlers did not differ in anthropometric indices, (ii) medallists had better results in generic fitness tests than non-medallists, (iii) medallists had better results in SWFT than non-medallists, and (iv) wrestlers did not differ in competition and training experience. 

### 4.1. No Differences in Anthropometric Indices between Medallists and Non-Medallists

Supportive to the results of our research, elite and amateur wrestlers from five different countries did not differ in body height, body mass, body mass index, and body fat percentage [9], and the same was noted among young Turkish wrestlers [7]. The main reason for these results could be found in the fact that wrestlers are competing in separate weight classes, which means that they are of a similar body type within the competitive category. Indeed, it was proven that body composition and body type depend on the wrestler’s weight category. For example, the endomorph–mesomorph somatotype dominates in the heavy-weight categories, while the balanced–mesomorph somatotype prevails in the lighter weight categories [18]. Therefore, as we included competitors from different categories, it is possible that both winners and defeated wrestlers from the same weight class would have similar anthropometric indices. Furthermore, it can be assumed that anthropometric indices are not that crucial for success in younger age categories, and some other factors are more important (e.g., technical, tactical skills, generic, and specific fitness performance). It should be noted here that also the static balance of the spine plays an important role in athletic performance [19]. 

### 4.2. Differences in Generic and Specific Performance between Medallists and Non-Medallists

Superior HGS is essential for the explosive execution of wrestling techniques and the initiation of attacks. It is important to note that HGS has been observed as one of the most important predictors of wrestling success [20]. Our results showed differences in HGS between medallists and non-medallists, which is in accordance with numerous previous studies. Specifically, it was recorded that in senior and junior Iranian Greco–Roman wrestlers, successful wrestlers had 8–18% greater HGS than less successful wrestlers [11]. Moreover, elite wrestlers produced up to 19% more strength in the handgrip compared to amateur wrestlers [9]. Similarly, Turkish junior national team members had higher HGS values than wrestlers not selected for the national team [21]. Özbay and Ulupınar [22] investigated whether the results in the HGS test, within a battery of strength-power tests, would be different when the tests are performed after exhaustive exercise in junior or under−23 athletes that belong to top-elite and elite wrestlers. The researchers showed that top-elite wrestlers presented higher relative results in all tests, except vertical and horizontal jump tests, when the tests were performed after exhaustive exercise. A note that top-elite wrestlers produced a higher output in both lower-body and upper-body Wingate average power (relative), compared to elite wrestlers, when the tests were performed after full rest [22]. 

Vertical jump performance differentiated successful from less successful young wrestlers in Croatia. It has previously been noted that elite wrestlers had higher values in the vertical jump test (CMJ) compared to amateurs (35.0 ± 3.5 and 31.9 ± 3.8 cm, respectively) [9]. Moreover, successful Iranian wrestlers had higher vertical jump values measured by the Sargent jump test than less successful wrestlers [11]. The observed differences in jumping capacities could be explained by better neuromuscular characteristics (i.e., neural activation patterns) among more successful wrestlers [9]. Explosive movements from the lower limbs are essential for performing successful lifting and throwing techniques that often decide the outcome of the match. Therefore, results indicate that wrestlers should work on their jumping capacities to enhance their overall performance.

Young wrestlers who were more successful in competitions also performed better in the specific wrestling test (i.e., SWFT). This finding is in accordance with several studies on older wrestlers that were published by the authors of the SWFT. More precisely, a study performed on Serbian wrestlers, aged 20–21, observed differences between wrestlers included in the first and second Wrestling League of Serbia, with first-league wrestlers having better results [23]. Moreover, adult wrestlers from three different competitive levels (national team wrestlers, first league, and second league wrestlers) differed in the SWFT performance, whereby higher-level wrestlers had better results (i.e., a higher number of throws) than lower-level wrestlers [15]. Therefore, according to the results of our study, it can be suggested that SWFT can be used for evaluating the specific physical fitness of young wrestlers, as it is sensitive to differentiating successful from less successful ones. This shows that this test can be a good indicator of a young wrestler’s fitness status and can follow their development of specific performance, which is crucial for success.

### 4.3. No Differences in Training and Competing Experience between Medallists and Non-Medallists

The finding that there were no differences in training and competing experience between medallists and non-medallists could seem surprising at first, but we will try to explain why this might have happened. The main reason lies in the fact that we included young wrestlers, who mostly start training harder and competing from the age of 10 years [24]. As authors, we are actively involved in the coaching practices for youth wrestlers, we will briefly try to describe these coaching practices.

Coaches try to stimulate young athletes with early competitions that are mainly low-level and friendly. Specifically, coaches go to competitions with children that have not been involved in wrestling training for a long time, even after just a few months of training. This way, coaches try to “break the routine” and provide children with more interesting situations alongside building a competitive nature of the sports training. Moreover, young wrestlers usually compete at competitions that are organized within or between clubs from the same city and county. This means that wrestling clubs organize frequent competitions on the community level, which are actually training sessions organized to look like a competition (i.e., low-level competition or simulation of the competition). Thus, the wrestlers we included in this study reported that they started to compete from a very early age and after a short training time (even after a couple of months of wrestling training), regardless of the level of the competition they participated in. Therefore, all wrestlers had similar (i.e., high) training and competing experience, as they were included in wrestling training and competitions from an early age, and that could be why we did not record differences in competition success and fitness performance.

### 4.4. Limitations and Strengths

The main limitation of this study is the cross-sectional type of research and the inability to make more precise conclusions. However, this study could serve as a guideline for future longitudinal and intervention studies. We can suggest that future studies conduct the same tests but on several testing points (e.g., at the beginning and after performing exhaustive exercise or after consecutive tests). Moreover, researchers could try to develop and apply specific training programs aimed to improve capacities that were identified as important in this study (i.e., CMJ, HGS, and SWFT; power, strength, anaerobic capacity) which would enable more accurate conclusions about whether those capacities are essential for becoming a successful wrestler (i.e., medallist). Furthermore, the strength of this research is that it included a respectful number of athletes of top and advanced quality, which is usually hard to collect and test. Furthermore, this study could have important implications for coaches as it provides good examples of testing athletes.

### 4.5. Practical Implications

This research has important practical implications. All tests included in this research assess different areas of the body and relate to various fitness parameters (power, strength, anaerobic capacity) that are essential for wrestling performance and are, most importantly, very easy and fast to conduct. Therefore, it could be suggested that coaches and practitioners include those tests during training and competition cycles to determine the performance status of their wrestlers. This way, coaches would be able to create optimal training programs that enable less successful wrestlers to develop the capacities essential to achieving better results and winning competitions. Moreover, coaches can use those tests to identify higher quality wrestlers and select them for the National team.

## 5. Conclusions

This study’s results noted that more successful wrestlers in competitions have better generic and specific sports performance, which means that selected tests can be used for fast and easy evaluation of a wrestler’s quality. Precisely, medallists had better results in the CMJ, HGS, and SWFT performance than non-medallists. Thus, these tests can be observed as very useful for determining the actual performance capacity of youth wrestlers. It should also be emphasized once more that these tests (CMJ, HGS, and SWFT) are very easy and fast to conduct, with minimal equipment, which makes them very feasible and appropriate for coaches to directly assess wrestlers’ performance capacities. Regarding the fact that coaches often have limited time to conduct the diagnostic testing of their athletes, identifying tests that are easy, fast, and sensitive for evaluating essential performance parameters is of great importance. Moreover, we propose that those tests become part of the universal fitness screening tests that all coaches in Croatia (or broader) should use. This would enable better connectedness of the coaches and National team selectors and lead to more precise and faster identification of successful wrestlers. This study identified which tests are able to differentiate successful from less-successful Croatian youth wrestlers, which has not been known previously. 

Wrestling coaches and scientists are constantly trying to learn what separates successful from less successful wrestlers to develop training plans that would allow all wrestlers to reach their full potential. The results from this study can help in creating optimal training plans, after conducting several tests as proposed in this study. Specifically, according to the results of this research, coaches should be oriented in creating training regimens that develop and enhance upper and lower limb strength and power and wrestling-specific strength and endurance. 

## Figures and Tables

**Figure 1 jfmk-08-00090-f001:**
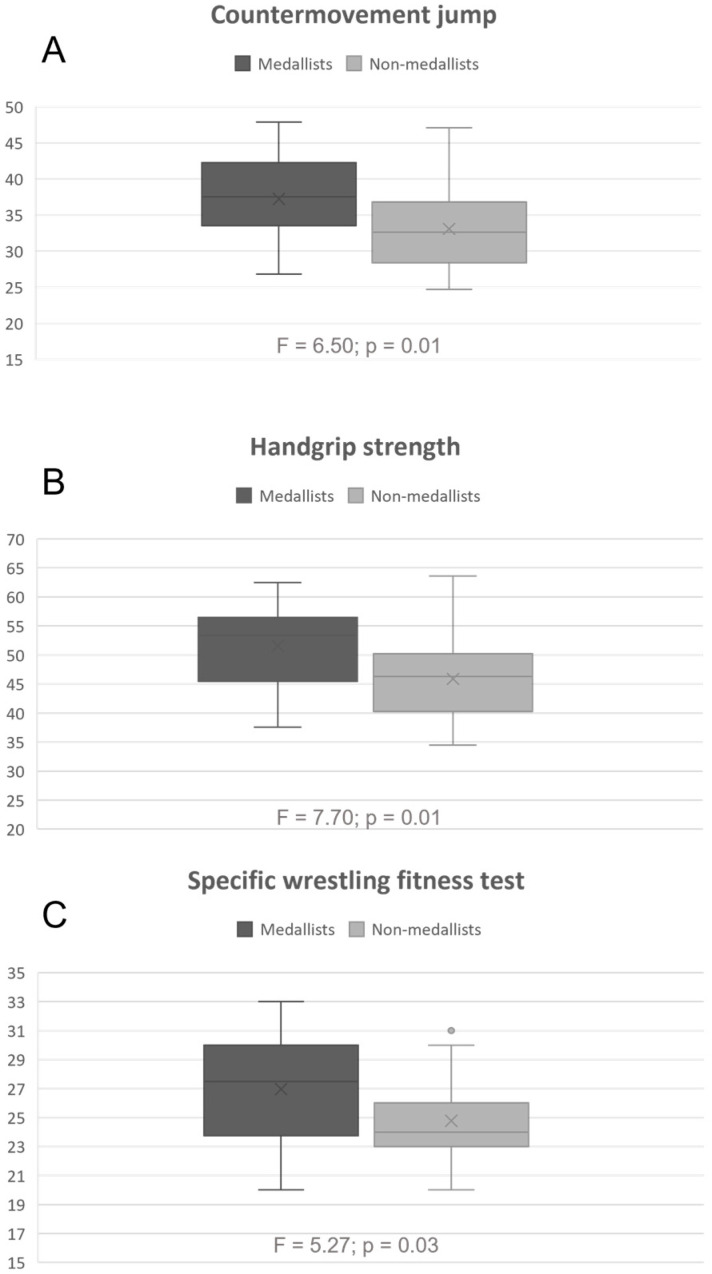
Graphical presentation of differences in the variables that significantly differentiate quality groups of wrestlers. (**A**) Countermovement jump; (**B**) Handgrip strength; (**C**) Specific Wrestling Fitness Test total repetitions.

**Table 1 jfmk-08-00090-t001:** Descriptive statistics and differences between medallists and non-medallists in all variables.

	Medallists (n = 26)		Non-Medallists (n = 23)		*t*-Test	
Variable	Mean	SD	Mean	SD	t-Value	*p*-Level
Age (years)	18	1.36	17.46	1.64	1.29	0.20
Training experience (years)	7.28	2.64	6.20	2.80	1.33	0.19
Competing experience (years)	6.80	2.69	5.80	2.97	1.18	0.24
Body weight (kg)	79.01	16.2	76.59	12.37	0.59	0.55
Body height (cm)	177.27	7.31	177.61	7.66	−0.16	0.87
Body mass index	24.93	3.52	24.17	2.89	0.84	0.40
Body fat percentage (%)	14.6	7.10	15.83	6.18	−0.66	0.51
Countermovement jump (cm)	37.22	5.75	33.08	5.6	2.55 *	0.01
Handgrip strength (kg)	51.55	7.22	45.87	7.06	2.77 *	0.01
SWFT total throws	26.96	3.61	24.78	2.95	2.30 *	0.03
SWFT total heart rate	347.11	19.93	342.5	18.16	0.60	0.55
SWFT index	12.92	2.19	13.98	1.83	−1.82	0.08

Note: n, number of subjects; SD, standard deviation; t, value of T-test; *p*, statistical significance; kg, kilograms; cm, centimeters; %, percentage; SWFT, Specific Wrestling Fitness Test, * denotes statistically significant differences.

**Table 2 jfmk-08-00090-t002:** Discriminant function analysis with function structure matrix according to performance categories.

Chi-Square Tests with Successive Roots Removed				
Eigenvalue	Canonical R	Wilks’ Lambda	Chi-Square	*p*-Value
0.37	0.52	0.73	13.2	0.04
**Factor Structure Matrix**				
Body weight		−0.19		
Body height		0.01		
Body fat percentage		0.16		
Countermovement jump		−0.64		
Handgrip strength		−0.69		
SWFT total throws		−0.59		

Note: SWFT, Specific Wrestling Fitness Test.

## Data Availability

Data for the current analysis are available upon request and can be obtained by contacting the corresponding author.

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
