# Peer review of "What Determines the Competitive Success of Young Croatian Wrestlers: Anthropometric Indices, Generic or Specific Fitness Performance?"

_jfmk, 2023, doi:10.3390/jfmk8030090_

Round 1
Reviewer 1 Report
This topic is very interesting, but some points need to be revised:
- Lines 85-87: "We hypothesize that more successful wrestlers would have favourable body indices and better fitness test results than their less successful peers." Improve this part. What is the purpose of this paper?
- Lines 168-170: Result section (table 1) shows that training experience is higher in medallists. Can this represent a bias in the study?
- Lines 207-209: "Also, it can be assumed that anthropometric indices are not that crucial for success in younger age categories, and some other factors are more important (e.g., technical, tactical skills, generic and specific fitness performance)." About the static balance of the spine of the athletes, very important papers should be considered. Look at these ones: -- doi: 10.1055/a-1962-0181 --- doi: 10.3390/jpm13050852 -- doi: 10.1007/s00586-022-07467-6 -- doi: 10.1093/eurheartj/ehad332
- Lines 260-265: "Coaches try to stimulate young athletes with early competitions that are mainly low-level and friendly. Moreover... of the level of the competition they participated in" This part is not clear. What do authors mean? Please revise it.
- Lines 289-290: "This study's results noted that more successful wrestlers in competitions have better generic and specific sports performance" Is this the primary objective of this paper? State here the most important results of this paper. What does this manuscript add new to the literature?
Minor editing of English language required.
Author Response
Jfmk revision
***REVIEWER 1***
This topic is very interesting, but some points need to be revised:
RESPONSE: Thank you for supporting our study and providing such valuable suggestions!
- Lines 85-87: "We hypothesize that more successful wrestlers would have favourable body indices and better fitness test results than their less successful peers." Improve this part. What is the purpose of this paper?
RESPONSE: Thank you for this suggestion. “We hypothesize that more successful wrestlers would have favourable body indices and better fitness test results than their less successful peers, which means that the selected tests and assessment tools would be able to differentiate wrestlers according to the performance quality. The results of this research would help clarify the factors that differentiate successful from less successful youth wrestlers and further determine which capacities are important for success in wrestling. Also, study could be used as a guideline for selecting tests and assessment tools that are appropriate and able to differentiate performance quality of youth wrestlers.”
- Lines 168-170: Result section (table 1) shows that training experience is higher in medallists. Can this represent a bias in the study?
RESPONSE: Yes, we are aware of that difference, even though it was not statistically significant. As we (especially the first author of the study) are actively involved in coaching youth athletes, we must say that we observed that one year more or less of training experience would not make a huge difference, knowing that wrestlers trained for approximately 6.5 years.
- Lines 207-209: "Also, it can be assumed that anthropometric indices are not that crucial for success in younger age categories, and some other factors are more important (e.g., technical, tactical skills, generic and specific fitness performance)." About the static balance of the spine of the athletes, very important papers should be considered. Look at these ones: -- doi: 10.1055/a-1962-0181 --- doi: 10.3390/jpm13050852 -- doi: 10.1007/s00586-022-07467-6 -- doi: 10.1093/eurheartj/ehad332
RESPONSE: Thank you for this suggestion, it is now briefly mentioned in the paper. Text now reads: “It should be noted here that also the static balance of the spine plays an important role in the athletic performance [19]”.
- Lines 260-265: "Coaches try to stimulate young athletes with early competitions that are mainly low-level and friendly. Moreover... of the level of the competition they participated in" This part is not clear. What do authors mean? Please revise it.
RESPONSE: It is now amended, and we tried to explain in more detail this part of the text. Text now reads: “Coaches try to stimulate young athletes with early competitions that are mainly low-level and friendly. Precisely, coaches go to competitions with children that have been involved in wrestling training not for a long time, even after just a few months of training. This way, coaches try to “break the routine” and provide children with more interesting situations alongside building a competitive nature of the sports training. Moreover, young wrestlers usually compete at competitions that are organized within or between clubs from the same city and county. This means that wrestling clubs organize frequent competitions on the community level, which are actually training sessions organized to look like a competition (i.e., low-level competition or simulation of the competition).”
- Lines 289-290: "This study's results noted that more successful wrestlers in competitions have better generic and specific sports performance" Is this the primary objective of this paper? State here the most important results of this paper. What does this manuscript add new to the literature?
RESPONSE: Thank you for noticing this gap in the explanation of the primary objective of the paper. One of the main objectives was to identify whether several selected tests can actually show the differences between performance quality of young wrestlers. It is now more explained in the Conclusion section. Text now reads: “This study's results noted that more successful wrestlers in competitions have better generic and specific sports performance, which means that selected tests can be used for fast and easy evaluation of wrestler’s quality. Precisely, medallists had better results in the CMJ, HGS, and SWFT performance than non-medallists. Thus, these tests can be observed as very useful for determining the actual performance capacity of youth wrestlers. It should also be emphasized once more that these tests (CMJ, HGS, and SWFT) are very easy and fast to conduct, with minimal equipment, which makes them very feasible and appropriate for coaches to directly assess wrestlers’ performance capacities. Regarding the fact that coaches often have limited time to conduct the diagnostic testing of their athletes, identifying tests that are easy, fast, and sensitive for evaluating essential performance parameters is of great importance. Moreover, we propose that those tests become a part of the universal fitness screening tests that all coaches in Croatia (or broader) should use. This would enable better connectedness of the coaches and National team selectors and lead to more precise and faster identification of successful wrestlers. This study identified which tests are able to differentiate successful from less-successful Croatian youth wrestlers, which has not been known previously.”
The study concerns the search for reasons for achieving better sports results by wrestlers of a similar age, characterized by similar training and competition experience in terms of anthropometric indicators, general fitness and special fitness.
RESPONSE: Thank you once again for your valuable comments and suggestions!
Reviewer 2 Report
The manuscript is well written, well structured, interesting, practical for coaches. It explores the importance of anthropometric indices, generic fitness and specific fitness in the performance of Croatian youth wrestlers.
The introduction provides enough background and adequately raises the issue of the study.
The methodology is well explained and the procedure description is properly detailed.
The results are synthetic, objective and adequate.
The discussion is deep and well structured, organized by the main findings of the study.
The authors are also aware of the limitations of the study, referring them appropriately at the end of the discussion.
Specific comments and suggestions:
· Lines 17-18: in the text appears “whether anthropometric indices and generic and specific fitness performance determine”. Suggestion: “whether anthropometric indices, generic and specific fitness performance determine”.
· Line 19: Despite being assumed, it should be mentioned that they are male subjects. The same comment for the sample description in the material and methods chapter.
· Lines 72-73: I suggest a comma, instead a dot after: “Markovic et al. 2022.”.
· Line 119: in the text appears “included handgrip strength (HGS)”. I suggest only “HGS”, since the full description already appears in the introduction. The some comment for “Handgrip strength” in the line 120.
· In table 1, I suggest that the decimal units be standardized, and that statistically significant differences are highlighted with "*", adding in the respective note the comparison it refers to.
· Line 173: in the text appears “It can be observed that that medallists”. Suggestion: “It can be observed that medallists”.
· Line 225: I suggest a comma after “jump tests,”.
· Lines 227-228: in the text appears “average power (relative), when compared with elite wrestlers when the tests were per-227 formed after full rest [21]”. Suggestion: “average power (relative), comparing to elite wrestlers, when the tests were performed after full rest [21]”.
Congratulations on the study!
Requires just a little bit of reading and correcting minor English issues.
Author Response
***REVIEWER 2***
The manuscript is well written, well structured, interesting, practical for coaches. It explores the importance of anthropometric indices, generic fitness and specific fitness in the performance of Croatian youth wrestlers.
The introduction provides enough background and adequately raises the issue of the study.
The methodology is well explained and the procedure description is properly detailed.
The results are synthetic, objective and adequate.
The discussion is deep and well structured, organized by the main findings of the study.
The authors are also aware of the limitations of the study, referring them appropriately at the end of the discussion.
RESPONSE: Thank you for recognizing the quality of our work, we highly appreciate it!
Specific comments and suggestions:
- Lines 17-18: in the text appears “whether anthropometric indices and generic and specific fitness performance determine”. Suggestion: “whether anthropometric indices, generic and specific fitness performance determine”.
RESPONSE: Thank you for noticing it. We corrected it, text now reads: “This research aimed to investigate whether anthropometric indices, and generic and specific fitness performance determine the competitive success of young wrestlers”.
- Line 19: Despite being assumed, it should be mentioned that they are male subjects. The same comment for the sample description in the material and methods chapter.
RESPONSE: It is now added in the text. Text reads: “This research included 49 Croatian Greco-Roman male wrestlers aged 17.75 ± 1.51 years”.
- Lines 72-73: I suggest a comma, instead a dot after: “Markovic et al. 2022.”.
RESPONSE: Corrected accordingly.
- Line 119: in the text appears “included handgrip strength (HGS)”. I suggest only “HGS”, since the full description already appears in the introduction. The some comment for “Handgrip strength” in the line 120.
RESPONSE: Thank you for noticing this mistake. It is now corrected. Text reads: “Generic fitness tests included HGS and the countermovement jump (CMJ). HGS was measured using the electronic hand dynamometer (Camry, Model EH101, Zhongshan Camry Electronic Co. Ltd. Kina).”
- In table 1, I suggest that the decimal units be standardized, and that statistically significant differences are highlighted with "*", adding in the respective note the comparison it refers to.
RESPONSE: Table 1 is now amended, please see Results section.
- Line 173: in the text appears “It can be observed that that medallists”. Suggestion: “It can be observed that medallists”.
RESPONSE: Thank you for noticing this typing mistake. It is now corrected.
- Line 225: I suggest a comma after “jump tests,”.
RESPONSE: It is now added. Text reads: “The researchers showed that top-elite wrestlers presented higher relative results in all tests, except vertical and horizontal jump tests, when the tests were performed after exhaustive exercise.”
- Lines 227-228: in the text appears “average power (relative), when compared with elite wrestlers when the tests were per-227 formed after full rest [21]”. Suggestion: “average power (relative), comparing to elite wrestlers, when the tests were performed after full rest [21]”.
RESPONSE: Amended accordingly. Text now reads: “A note that top-elite wrestlers produced a higher output in both lower-body and upper-body Wingate average power (relative), comparing to elite wrestlers, when the tests were performed after full rest[21]”.
Congratulations on the study!
RESPONSE: Thank you once again!
Reviewer 3 Report
The study concerns the search for reasons for achieving better sports results by wrestlers of a similar age, characterized by similar training and competition experience in terms of anthropometric indicators, general fitness and special fitness.
The authors wanted to discover which of the studied factors were more significant in achieving sports success.
For this purpose, they correctly selected and carried out anthropometric measurements, as well as conducted appropriate (specific) tests of general and special fitness for the discipline. The methodology of the applied statistical calculations presented in the results also raises no objections.
The discussion and conclusions were well structured.
The advantage of the study is its substantive reliability, and above all, the presentation by a group of wrestling experts - scientists of proposals for the trainers to use the presented methods of assessing sports preparation, which can be used at various stages of sports training. Another advantage of this study is the enrichment of the poor scientific literature on the subject for a specific sports group, which are wrestlers.
The publication meets all the requirements for such studies. I encourage the actors to continue research and look for connections with the assessment of the training status of players in similar disciplines, e.g. judo, where similar methods of measuring general and special fitness could be used by coaches of these disciplines in shaping outstanding athletes.
Author Response
***REVIEWER 3***
The authors wanted to discover which of the studied factors were more significant in achieving sports success.
For this purpose, they correctly selected and carried out anthropometric measurements, as well as conducted appropriate (specific) tests of general and special fitness for the discipline. The methodology of the applied statistical calculations presented in the results also raises no objections.
The discussion and conclusions were well structured.
The advantage of the study is its substantive reliability, and above all, the presentation by a group of wrestling experts - scientists of proposals for the trainers to use the presented methods of assessing sports preparation, which can be used at various stages of sports training. Another advantage of this study is the enrichment of the poor scientific literature on the subject for a specific sports group, which are wrestlers.
The publication meets all the requirements for such studies. I encourage the actors to continue research and look for connections with the assessment of the training status of players in similar disciplines, e.g. judo, where similar methods of measuring general and special fitness could be used by coaches of these disciplines in shaping outstanding athletes.
RESPONSE: Thank you for recognizing the quality and importance of our study. These comments motivated us to continue similar investigations even more!
Round 2
Reviewer 1 Report
Well done.
Minor editing of English language required